# The Screening of microRNAs in Chronic Myeloid Leukemia: A Clinical Evaluation

**DOI:** 10.3390/ijms25063363

**Published:** 2024-03-16

**Authors:** Denise Kusma Wosniaki, Anelis Maria Marin, Rafaela Noga Oliveira, Gabriela Marino Koerich, Eduardo Cilião Munhoz, João Samuel de Holanda Farias, Miriam Perlingeiro Beltrame, Dalila Luciola Zanette, Mateus Nóbrega Aoki

**Affiliations:** 1Laboratory for Applied Science and Technology in Health, Carlos Chagas Institute, Oswaldo Cruz Foundation (Fiocruz), Curitiba 81350-010, Brazil; anelis.marin@fiocruz.br (A.M.M.); rafaelanogaoliveira@gmail.com (R.N.O.); gabriellakoerich@gmail.com (G.M.K.); dalila.zanette@fiocruz.br (D.L.Z.); 2Erasto Gaertner Hospital, Curitiba 81520-060, Brazil; edciliao@hotmail.com (E.C.M.); jshfarias@erastogaertner.com.br (J.S.d.H.F.); citometria@erastogaertner.com.br (M.P.B.)

**Keywords:** chronic myeloid leukemia, microRNA, imatinib, *BCR::ABL1*, biomarkers

## Abstract

Chronic myeloid leukemia (CML) is a type of leukemia whose main genetic marker is the reciprocal translocation that leads to the production of the *BCR::ABL1* oncoprotein. The expression of some genes may interfere with the progression and development of leukemias. MicroRNAs are small non-coding RNAs that have the potential to alter the expression of some genes and may be correlated with some types of leukemia and could be used as biomarkers in the diagnosis and prognosis of patients. Therefore, this project carried out an analysis of microRNA-type plasma biomarkers in patients with chronic myeloid leukemia at unique points, including follow-up analysis of patients from the Erasto Gaertner Hospital. 35 microRNAs were analyzed in different cohorts. Inside those groups, 70 samples were analyzed at unique points and 11 patients in a follow-up analysis. Statistically different results were found for microRNA-7-5p, which was found to be upregulated in patients with high expression of the *BCR::ABL1* transcript when compared to healthy controls. This microRNA also had evidence of behavior related to *BCR::ABL1* when analyzed in follow-up, but strong evidence was not found. In this way, this work obtained results that may lead to manifestations of a relationship between miR-7-5p and chronic myeloid leukemia, and evaluations of possible microRNAs that are not related to this pathology.

## 1. Introduction

Chronic myeloid leukemia is a malignant myeloproliferative disease, mainly caused by a reciprocal translocation between chromosomes 9 and 22 (9; 22) (q34; q11), resulting in the Philadelphia chromosome (Ph), and the corresponding *BCR::ABL1* fusion gene, which is the main biomarker of this pathology [1,2].

The resulting transcriptional fusion encodes the oncoprotein *BCR::ABL1*, which is a constitutively active tyrosine kinase, whose activity stimulates leukemogenesis [3]. Therefore, treatment is carried out with tyrosine kinase inhibitors (TKIs), and despite this being a highly efficient treatment, approximately 13% of patients develop resistance to this class of compounds [4].

Patient monitoring through molecular quantification of the *BCR::ABL1* transcript is highly relevant for evaluating the prognosis of patients with CML. Treatment failure is related to mechanisms of resistance to TKIs, the origin of which may be a mutation in the coding region of the kinase domain of the *BCR::ABL1* tyrosine kinase. Several mutations in this domain have already been described, such as T315I, V299L, G250E, F317L, Y253H, E255K/V, F359V/C/I, and L248V [5] among others. Although CML has *BCR::ABL1* as both a diagnostic and monitoring marker, the search for other plasma biomarkers are important for the prognosis of the disease. Among the biomolecules that can be explored with this potential are non-coding RNAs (ncRNAs), which can help in the identification of altered or modulated targets in different phases of the disease; this can be extremely relevant in clinical use, as they can be related to resistance to the chemotherapy drugs used.

MicroRNAs make up a broad and well-studied class of ncRNAs, being correlated with many signaling pathways [6] playing a role in the regulation of gene expression, RNA maturation, protein synthesis, and can also have their activity regulated at a post-transcriptional level [7,8,9].

Considering the presented information, this study aimed to evaluate the dynamics of microRNA expression in samples from patients with CML, correlating the results with the stage of the disease and *BCR::ABL1* transcript quantification to help in the disease follow-up and prognosis.

## 2. Results

### 2.1. Epidemiological Description

The epidemiological data of participants recruited for single-point analyses (patients with a single sample collected) are described in Table 1.

### 2.2. Single-Point Plasma Analysis

In the first stage, eight microRNAs were evaluated: miR-17-5p [10,11,12,13], miR-23a-3p [14,15,16], miR-93-5p [17,18], miR-130a-3p [19], miR-142-5p [20,21], miR-148b-3p [22,23], miR-199a-3p [24,25], and miR-331-5p [26]. These microRNAs have been chosen based on oncology potential found in the literature. Some of them are related to hematological malignancies and others are related to other types of cancer, in addition to the exogenous normalizer cel-miR-39-3p. For statistical analysis, the samples were subdivided into three sample groups: samples with a value higher than 0.1% of *BCR::ABL1* transcript, samples with undetectable *BCR::ABL1,* and samples from healthy individuals (Table 2). No significant differences were observed in the expression of these eight miRNAs between the three sample groups (considering a *p* value of <0.05).

In the second stage, 27 additional microRNAs were evaluated, including 10 samples with a high percentage of *BCR::ABL1* transcript (0.1% or more) and 10 samples from the healthy control group. The miRNAs evaluated were: miR-7-5p, miR-19b-3p, miR-20a-5p, miR-21-5p, miR-25-3p, miR-27b-3p, miR-29a-3p, miR-29b-3p, miR-92a-1-5p, miR-103a-3p, miR-106b-5p, miR-122-5p, miR-125b-5p, miR-130a-5p, miR-150-5p, miR-155-5p, miR-186-5p, miR-192-5p, miR-193b-3p, miR-205-5p, miR-214-5p, miR-221-3p, miR-221-5p, miR-361-5p, miR-451, miR-486-5p, and miR-494-3p (Table 3).

There was a significantly higher expression of miR-92a-1-5p in samples from patients with elevated percentages of the *BCR::ABL1* transcripts, compared to the healthy control group (*p* = 0.0220) (Figure 1), in spite of great variation among CML samples. The other microRNAs tested did not show significant differential expressions between the sample groups.

Among the 27 microRNAs tested, miR-92a-1-5p and three other miRNAs were analyzed in a greater number of samples in the groups (*n* = 20): miR-92a-1-5p, miR-7-5p, miR-486-5p, and miR-361-5p (Table 4). These three other miRNAs were those with a tendency for significant results between groups in the first 10 samples.

There was a significantly higher expression of miR-7-5p in samples from patients with a higher *BCR::ABL1* proportion, when compared to the control group (*p* = 0.0353) (Figure 2). The microRNAs miR-486-5p and miR-361-5p did not show a significant difference in expression between the tested groups. When we increased the number of samples, miR-92a-1-5p had no significant difference in expression.

For the remaining analyses, only miR-7-5p was used for the evaluation of all samples from patients with expressions of the *BCR::ABL1* transcript (30 samples) and in the group of samples from CML patients with undetectable levels of *BCR::ABL1* transcripts (20 samples), as well as samples from healthy controls (Figure 3).

There was a significant difference in expression between groups with a high percentage of *BCR::ABL1* and healthy controls (*p* = 0.0342) (Figure 3). There was no significant difference between the groups with a high percentage of *BCR::ABL1* and the group with undetected *BCR::ABL1* (Figure 3).

### 2.3. Buffy Coat Single-Point Analysis

For expression analysis with RNA extracted from the buffy coat, the endogenous miUSB U6 was used as a normalizer [27,28]. The analyses were carried out mostly with the same patients that were used for the plasma analyses. However, some of the buffy coat samples were not available, so the final number of samples is not the same.

Thirteen samples from patients with a high percentage of *BCR::ABL1*, nine samples from patients with undetected *BCR::ABL1*, and five samples from the healthy control group were analyzed. miR-7-5p was evaluated in RNA from buffy coats with the aim of corroborating the results seen in plasma RNA. However, as opposed to plasma, no significant differences were seen in the expression of miR-7-5p in buffy coat RNA (Table 5 and Figure 4).

### 2.4. Follow-Up Analysis

This analysis was carried out with miR-7-5p, and for this purpose, we selected patients that had two or more sample points collected, and subsequently divided them into two groups: those under treatment and those in treatment-free remission (TFR). The objective of this analysis was to evaluate a possible relationship between the expression of miR-7-5p and treatment interruption or failure, considering mainly the quantification of the *BCR::ABL1* transcript as a reference parameter.

Patients undergoing follow-up were divided into two groups: patients undergoing treatment (initially with Imatinib) and patients who underwent treatment interruption (treatment-free remission); six patients were selected for the first group and four patients for the second group. The patients selected for these analyses may also be within one of the single-point analysis groups, and for this analysis, these patients were selected individually to carry out follow-up monitoring.

#### 2.4.1. Group 1: Patients Undergoing Treatment

Regarding patients undergoing treatment, patients who started their treatment with Imatinib and had distinct progressions based on *BCR::ABL1* levels were selected. A correlation was analyzed using Sperman’s rank correlation coefficient and was not found for any of the patients evaluated, despite some patients who showed a concordant relationship between *BCR::ABL1* and miR-7 levels. However, in data from other patients, this relationship was not observed, and this may have happened due to other factors which we do not know, and may be interfering with the expression of this microRNA and not in the expression of the *BCR::ABL1* transcript (Figure 5 and Table 6).

In this context, the results of patient 1 (Table 6) at the fourth point show there was a new increase in the expression of both microRNA and the percentage of leukemic cells. However, at the fifth point, *BCR::ABL1* levels decreased and microRNA expression increased. This patient had a failure in their treatment with Imatinib, and as we noted in the last point, the treatment was changed to Dasatinib. Therefore, the microRNA followed the expression levels of the *BCR::ABL1* transcript and the last point may be related to the predisposition to instability or treatment failure. However, this hypothesis could only be deepened by analyzing subsequent samples from the same patient.

Patient 2 (Table 6) had a good treatment progression with Imatinib, since this patient’s results expressed a decreased percentage of *BCR::ABL1*, the expression of miR-7-5p continued showing low levels. Their data were consistent.

Regarding patient 3 (Table 6), we observed instability in the treatment with Imatinib, which may have been generated by a lack of correct adherence to the treatment. There is a fluctuation in the value of *BCR::ABL1*, which is accompanied by an increase in the expression of miR-7-5p, indicating that the greater the expression of miR-7-5p, the greater the possibility of worsening or treatment failure. At the last point for this patient, there was a decrease in the expression of miR-7-5p and in the percentage of the *BCR::ABL1* transcript.

According to the results of patient 4 (Table 6), this also showed instability in the treatment, which is demonstrated by the increase in miR-7-5p expression levels at the third point, where *BCR::ABL1* was already high. However, at the fourth point, there was a change in treatment to Dasatinib, and microRNA expression levels decreased, along with *BCR::ABL1* levels. At point 5, both increased again, which could be a warning sign that the treatment may not be working. This hypothesis could only be supported when related to subsequent analyses of the same patient.

The results of patient 5 (Table 6) showed a reduction in *BCR::ABL1* levels from points 1 to 5, where miR-7-5p levels were also reduced compared to the baseline. However, the levels of *BCR::ABL1* at point 6 increased in relation to the previous point, while the level of miR-7-5p was already increased at the previous point, indicating that this miRNA may have had an increase prior to the elevation of *BCR::ABL1*, and, consequently, suggesting a relationship between micrRNA-7-5p and the increase in *BCR::ABL1*.

The sixth patient (Table 6) showed a good progression in treatment when we evaluated the expression of *BCR::ABL1*, but the expression of microRNA 7-5p remained unstable throughout the period, which could be a warning sign of possible future treatment failure.

#### 2.4.2. Patients in Treatment-Free Remission

The patients evaluated in these analyses were selected by professionals at the Erasto Gaertner Hospital to begin treatment-free remission. The criteria for selecting these patients were evaluated according to the NCCN guidelines (NCCN Clinical Practice Guidelines in Oncology (NCCN Guidelines^®^) [29], Chronic Myeloid Leukemia, Version 2.2024—5 December 2023). Regarding the results obtained in the analyses of the five selected patients in free remission (Table 7), it was observed that in patient 1 there were two follow-up points where the expression levels of miR-7-5p were increased, while the levels of *BCR::ABL1* remained undetectable. In patient 2, the expression of miR-7-5p remained low at all points. The third patient had stable and low microRNA expression at all points in their free remission. Finally, the fourth patient had a long follow-up, and a point of instability was observed, with an increase in the expression of miR-7-5p (point 8), returning to stability shortly thereafter.

## 3. Discussion

Several studies have shown that MicroRNAs are associated with different biological pathways and have a great influence on the transduction of cell signaling [30]. The involvement of microRNAs in the development, progression, and resistance to chemotherapy in CML [31] shows that they can be important targets for studies related to chronic myeloid leukemia and other types of cancer. The present study addressed different clinical cohorts, forming different points of analysis aiming to evaluate possible biomarkers.

To achieve this, a screening of 35 microRNAs was first carried out in plasma samples from a cohort of patients with CML. These microRNAs were selected according to previous data in the literature, with some being correlated to leukemias in different contexts, such as diagnosis, prognostic assessment, and Treatment-Free Remission protocol, while some were selected based on their scientific relevance in other types of cancer. The results obtained showed that miR-7-5p had a significant differential expression among groups of patients separated according to their percentage of *BCR::ABL1* expression.

The expression of miR-7-5p was significantly increased in samples from patients with a high percentage of *BCR::ABL1* transcript compared to healthy controls. Surprisingly, as observed in the results, this microRNA had the highest expression in some CML patients treated with Imatinib and with a high percentage of *BCR::ABL1*, while other patients in this same group presented values similar to the average of controls. This result suggested that microRNA 7-5p may be related to the progression of the disease, considering that the increase in its expression could be related to the increase in the percentage of *BCR::ABL1* transcript. These results led us to perform two additional analyses: single point analysis with RNA samples obtained from leukocytes (buffy coat) from the same patients, and analysis at various follow-up points of patients with CML.

To identify the origin of miR-7-5p in plasma, we evaluated its expression in leukocytes from the same patients. Unlike what was observed in plasma, miR-7-5p did not have a differential expression in the patients’ leukocytes. The increase of miR-7-5p in plasma could be related to the release of extracellular vesicles containing microRNA-7-5p or other factors, such as the hypothesis that this microRNA is being secreted by cells of the bone marrow and not directly from leukocytes in the bloodstream.

As another approach to investigate the role of miR-7-5p in CML, we evaluated a follow-up cohort of samples from the same patients collected at consecutive points throughout their progression with Imatinib treatment. Similar to what was found in the previous stage, miR-7-5p showed different behaviors in relation to the course of the disease in each patient, indicating a heterogeneous behavior of this target, suggesting that it can be modulated and altered by several factors.

These results indicate that microRNA-7-5p may be related to the expression of the *BCR::ABL1* transcript. Some points of the analysis showed sudden increases in the expression of miR-7-5p, indicating that it can be a monitoring biomarker. Furthermore, it was also observed that microRNA-7 had a coincident progression relationship with the *BCR::ABL1* transcript, where both increased or decreased at the same time, which may be related only to the *BCR::ABL1* transcript and not with the prediction of prognosis. However, monitoring more patients for longer periods of time is necessary to confirm this hypothesis.

Another approach of the present study evaluated patients who started treatment with Imatinib, and, as they had excellent progression, were enrolled for treatment-free remission. All of these patients had undetectable *BCR::ABL1* t at all follow-up points. In this context, the molecular monitoring of *BCR::ABL1* expression in TFR patients together with the plasma quantification of miR-7-5p aimed to observe whether miR-7-5p could serve as an anticipatory marker of increased expression of *BCR::ABL1*, indicating the need to resume discontinuation for these patients.

Of the 4 TFR patients, 3 maintained stable miR-7-5p expression at all points of their follow-up, indicating that there may be some type of relationship with the worsening prognosis and the increase in the percentage of leukemic cells. Two patients had unstable results, such as patient 1 who had an instability where they had an increase in the expression of miR-7-5p in the third and fourth points. Considering the instability in patient 5, it can be hypothesized that there are other factors interfering in miR-7-5p expression. More follow-up points would be necessary so that we can draw any conclusions.

MicroRNA-7 has been studied in several types of cancer [32], such as lung, hepatocellular, breast, gliomas, colorectal, hematological neoplasms, among others [33]. This microRNA has been characterized as a tumor suppressor in several types of cancer [34] and has also been related to the modulation of signaling pathways [35,36]. MicroRNA-7 may have a functional performance related to the inhibition of DNA repair mechanisms, carried out by PARP-1 and BRCA [37]. A study indicated that there is a relationship between ANRIL/miR-7 in which microRNA-7 may function as a tumor suppressor in T-cell acute lymphoblastic leukemia [32]. A possible miRNA-TET2 pathway was also identified, where microRNAs, including miR-29b, miR-101, miR-125b, miR-29c, and miR-7 are overexpressed and thus may be involved in the pathogenesis of AML [38].

In the context of chronic myeloid leukemia, the roles for miR-7 are not yet well understood. Jiang et al. (2017) carried out studies demonstrating that miR-7 inhibited cell proliferation and promoted apoptosis in K562 cell lines focusing on the *BCR::ABL1/PI3K/AKT* signaling pathway [39]. The same study demonstrated that microRNA-7 may also be related to the sensitization of the K562 cell to Imatinib [39]. Considering the studies already carried out and the results obtained in this work, we can hypothesize that these pathways may be related to changes in the expression of microRNA-7/*BCR::ABL1*.

There is little evidence about the relationship between microRNA-7-5p and chronic myeloid leukemia patients. However, the present work found evidence that miR-7-5p may be related to chronic myeloid leukemia in a non-specific way, mainly considering its relationship with the *BCR::ABL1* transcript. It is also important to highlight that the modern study of oncology, mainly focused on diagnosis, prognosis, and therapeutic evolution, is linked to individual differences and patient profiles. Therefore, the identification of a single sensitive and specific biomarker for a type of neoplasm is unlikely, requiring studies with a panel of biomarkers and the stratification of patients with a given neoplasm to elucidate more sensitive and specific biomarkers. This reiterates the importance of microRNAs as excellent accessories and mainly individualized biomarkers, being important in precision medicine, which is the main objective of clinical oncology, as it directs efforts, resources, time, and patient survival, which increases their quality of life.

## 4. Materials and Methods

### 4.1. Casuistry

Peripheral blood samples from patients with CML were collected at Erasto Gaertner Hospital (Curitiba, PR, Brazil), for research projects approved according to CAAE 08809419.0.0000.0098 and 53207021.5.0000.0098. The microRNAs addressed in this project were previously selected based on studies related to microRNAs in CML, consisting of 35 miRNAs: miR-7-5p, miR-17-5p, miR-19b-3p, miR-20a-5p, miR-21-5p, miR-23a-3p, miR-25-3p, miR-27b-3p, miR-29a-3p, miR-29b-3p, miR-92a-1-5p, miR-93-5p, miR-103a-3p, miR-106b-5p, miR-122-5p, miR-125b-5p, miR-130a-3p, miR-130a-5p, miR-142-5p, miR-148b-3p, miR-150-5p, miR-155-5p, miR-186-5p, miR-192-5p, miR-193b-3p, miR-199a-3p, miR-205-5p, miR-214-5p, miR-221-3p, miR-221-5p, miR-331-5p, miR-361-5p, miR-451, miR-486-5p, and miR-494-3p. The miRNAs were evaluated in cohorts of CML patients, who were divided into groups and selected considering the following variables: age, sex, type of leukemia, time of diagnosis, and type and duration of treatment. Thus, the cohorts analyzed at single-point samples were: (1) High, samples from Imatinib-treated patients, with *BCR::ABL1* > 0.1% (*n* = 30); (2) ND, samples from Imatinib-treated patients, with *BCR::ABL1* < 0.1% or undetectable (*n* = 20); (3) Healthy Control (HC) samples from healthy blood donors (*n* = 20); and a fourth group of serial samples; and (4) Follow-up cohort: samples from 11 patients that contained several collection points, at different points in the treatment, to evaluate the evolution of the disease through the quantification of *BCR::ABL1* by RT-qPCR.

### 4.2. Buffy Coat RNA Extraction

After plasma separation by centrifugation, the buffy coat was subjected to total RNA extraction using the QIAamp^®^ RNA Blood Mini Kit (Qiagen, Hilden, Germany). The extracted RNA was quantified in NanoDrop™ One (Thermo Fisher Scientific, Waltham, MA, USA) and stored at −80 °C until use.

### 4.3. Quantification of BCR::ABL1

Total RNA extracted from the buffy coat was analyzed by RT-qPCR according to the protocol described by Marin et al., 2023 [40].

### 4.4. Synthesis of Endogenous cDNA from Buffy Coat Samples

After extracting the RNA from the buffy coat, this RNA was used to quantify the *BCR::ABL1* transcript, as described in Marin et al., 2023, and for the quantification of microRNAs. The endogenous miUSB-U6 was used to quantify microRNAs, and cDNA synthesis was performed using the High-Capacity cDNA Reverse Transcription Kit (Thermo Fisher Scientific), following the manufacturer’s instructions.

### 4.5. Extraction of Plasma RNA

Before starting the extraction, 1 µL of cel-miR-39-3p template at 1 nM (Spike-in-control) was added to each 100 µL of plasma. Then the total RNA was extracted using the Magmax^TM^ mirVana^TM^ Total RNA Isolation Kit (Thermo Fisher Scientific), following the instructions for use for RNA isolation from serum and plasma samples.

### 4.6. cDNA Synthesis of Free MicroRNAs in Plasma

For reverse transcription, the TaqMan^TM^ Advanced miRNA cDNA Synthesis kit (Thermo Fisher Scientific) was used, following the manufacturer’s instructions.

### 4.7. Real-Time PCR of microRNAs

For the relative quantification of miRNAs, assays were used with TaqMan probes specific for each microRNA (Thermo Fisher Scientific) and the TaqMan™ Fast Advanced Master Mix (Thermo Fisher Scientific). The PCR reaction was carried out on a QuantStudio 5^TM^ real-time PCR platform (Thermo Fisher Scientific) using the exogenous microRNA cel-miR-39-3p as a normalizer. The qPCRs were performed following the manufacturer’s protocol, in duplicate for each sample and always using a negative control (the no template control, NTC).

### 4.8. Statistical Analyses

The data were calculated using the 2^−ΔΔCT^ methodology [41] where the results obtained for single-point analyses were performed using the Graphpad Prism 7 software. Expression data was normalized with the median of the Ct values of endogenous control miUSB-U6 and the exogenous control cell- miR-39-3p.

## 5. Conclusions

35 microRNAs were evaluated, namely: miR-7-5p, miR-17-5p, miR-19b-3p, miR-20a-5p, miR-21-5p, miR-23a-3p, miR-25-3p, miR-27b-3p, miR-29a-3p, miR-29b-3p, miR-92a-1-5p, miR-93-5p, miR-103a-3p, miR-106b-5p, miR-122-5p, miR-125b-5p, miR-130a-3p, miR-130a-5p, miR-142-5p, miR-148b-3p, miR-150-5p, miR-155-5p, miR-186-5p, miR-192-5p, miR-193b-3p, miR-199a-3p, miR-205-5p, miR-214-5p, miR-221-3p, miR-221-5p, miR-331-5p, miR-361-5p, miR-451, miR-486-5p, and miR-494-3p. This work showed that the majority of microRNAS analyzed do not have a relationship with chronic myeloid leukemia. However, it was also observed that microRNA 7-5p may have a relationship with chronic myeloid leukemia and the *BCR::ABL1* transcript in different ways.

## Figures and Tables

**Figure 1 ijms-25-03363-f001:**
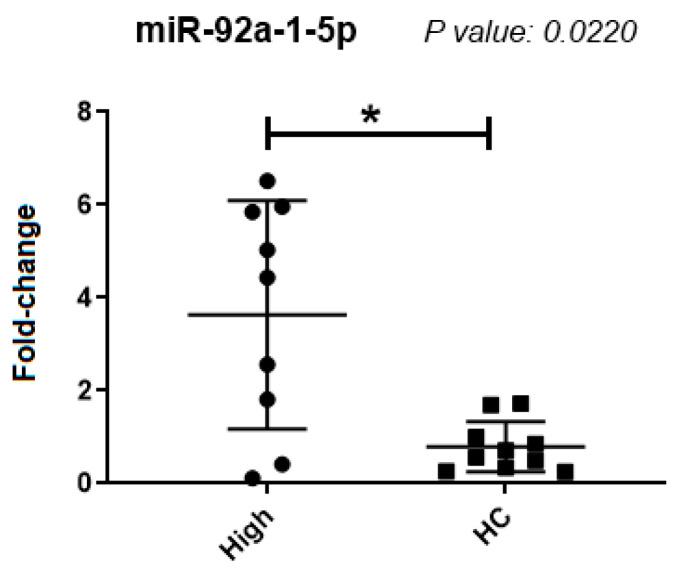
miR-92a-1-5p expression. Comparative results in a group of samples from patients with CML and healthy donors. High: samples with a high percentage of leukemic cells; HC (Healthy Controls): Samples from healthy donors. *: Significant difference (*p* < 0.05).

**Figure 2 ijms-25-03363-f002:**
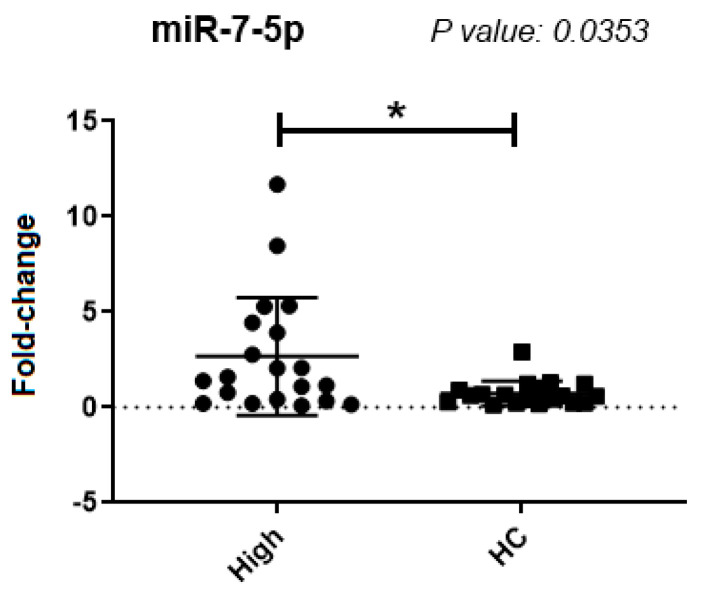
miR-7-5p expression analysis with additional samples. High: samples with a high percentage of leukemic cells; HC (Healthy Controls): Samples from healthy donors. *: Significant difference (*p* < 0.05).

**Figure 3 ijms-25-03363-f003:**
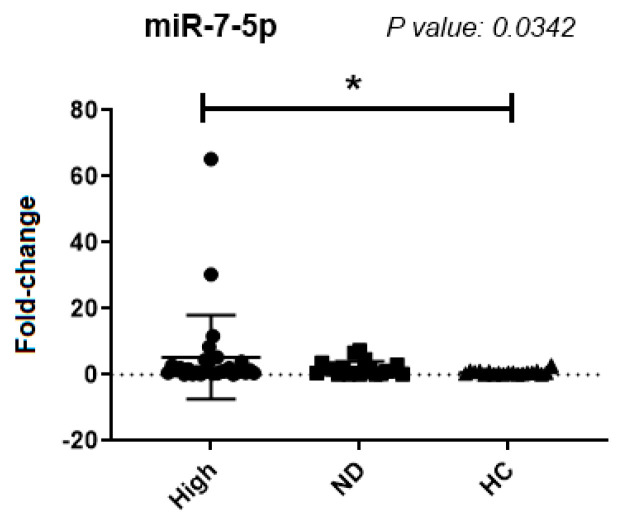
miR-7-5p expression analysis between the group with a high percentage of *BCR::ABL1*, group with undetectable *BCR::ABL1,* and samples from healthy donors. High: samples with a high percentage of leukemic cells; ND (Non-Detected): Samples with an undetectable percentage of leukemic cells; HC (Healthy Controls): Samples from healthy donors. Mean ± deviation: High: 2.6 ± 3.1; ND: 1.8 ± 2.2 and HC: 9.5 ± 2.2. *: significant difference (*p* < 0.05).

**Figure 4 ijms-25-03363-f004:**
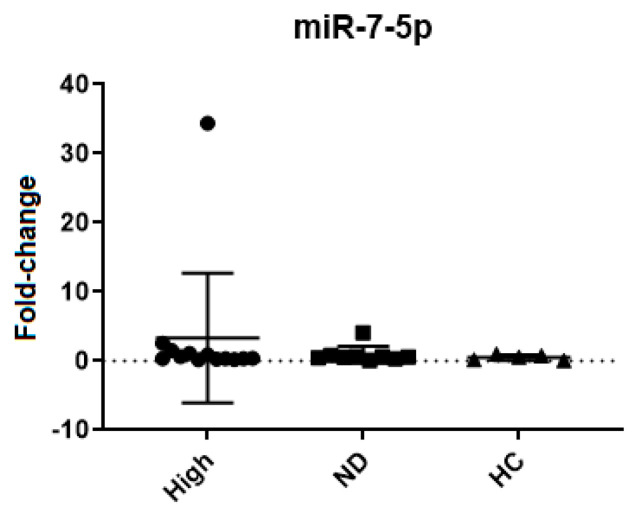
miR-7-5p expression analysis in RNA from buffy coat, the results shown non-significant (ns) statistics evaluating this group (data shown in Table 5). Groups: *BCR::ABL1* high; undetectable *BCR::ABL1,* and healthy donors; High: samples with a high percentage of leukemic cells; ND (Non-Detected): Samples with an undetectable percentage of leukemic cells; HC (Healthy Controls): Samples from healthy donors.

**Figure 5 ijms-25-03363-f005:**
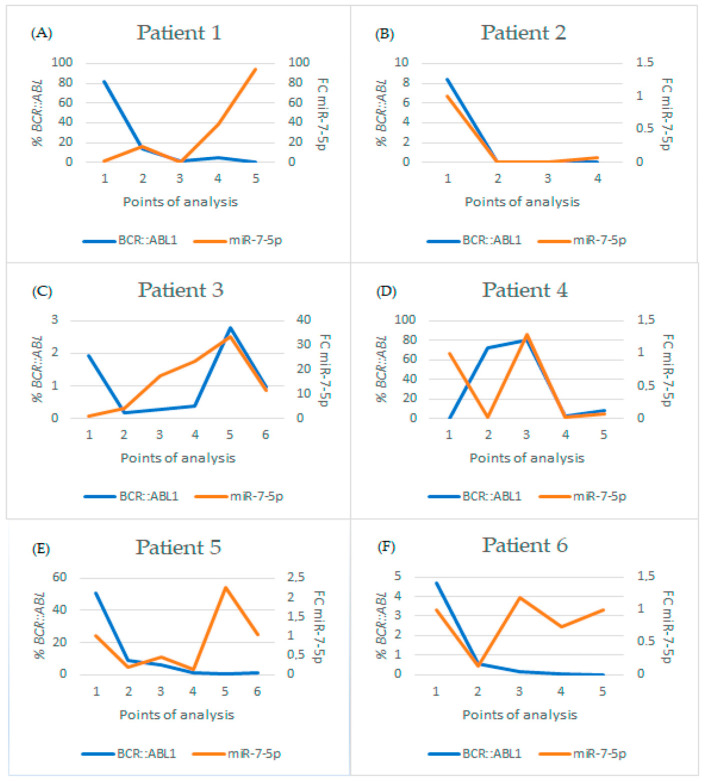
Graphs showing the analysis of patients undergoing continuous follow-up to evaluate the progression of the pathology by evaluating the expression of microRNA-7-5p (fold-change, left scale, and orange line), expression of the *BCR::ABL1* transcript (percentage, right scale, and blue line), treatment and treatment time, evaluating the date that is shown in Table 6. (**A**) Patient 1; (**B**) Patient 2; (**C**) Patient 3; (**D**) Patient 4; (**E**) Patient 5; and (**F**) Patient 6. *BCR::ABL1*: Percentage of *BCR::ABL1* transcript in the sample; FC: Fold Change.

**Table 1 ijms-25-03363-t001:** Epidemiological data of patients recruited for single-point analyses and *p* value related to Kruskall–Wallis analysis showing the significances of differences between the ages of the respective groups. High: 0.1% or higher percentage of *BCR::ABL1* transcript; ND: Non Detected *BCR::ABL1* transcript; HC: Health Control group. ns: non-significant.

Group	*BCR::ABL1*	Sex		Treatment Time in Months (Mean ± Deviation)	Age (Mean ± Deviation)	Age Differences between Groups (*p* Value and Significance)		
		M	F			High vs. ND	High vs. HC	ND vs. HC
1	High	18	12	15.3 ± 29.4	50.4 ± 20.7	0.1291	>0.9999	0.1606
2	ND	10	10	54 ± 23.6	62 ± 13.4	ns	ns	ns
3	HC	10	10	-	51.8 ± 6.9	-	-	-

**Table 2 ijms-25-03363-t002:** *p* value related to Kruskall–Wallis analysis showing the general rank *p* value and the *p* value for multiple comparisons. High: 0.1% or higher percentage of *BCR::ABL1* transcript; ND: Non Detected *BCR::ABL1* transcript; HC: Health Control group.

miRNA	General Rank	Multiple Comparisons (*p* Value)			Mean ± Deviation		
		High vs. ND	High vs. HC	ND vs. HC	High	ND	HC
miR-93-5p	0.9979	>0.9999	>0.9999	>0.9999	12.1 ± 37.1	1.8 ± 2.2	5.6 ± 18.8
miR-23a-3p	0.8934	>0.9999	>0.9999	>0.9999	9.3 ± 28.1	1.1 ± 1.6	1.2 ± 2.6
miR-199a-3p	0.9534	>0.9999	>0.9999	>0.9999	12.2 ± 38.6	1.9 ± 2.6	3.1 ± 8.05
miR-17-5p	0.9005	>0.9999	>0.9999	>0.9999	1.5 ± 3.5	0.4 ± 0.8	0.6 ± 1.7
miR-148b-3p	0.5818	>0.9999	0.9747	>0.9999	19.8 ± 54.4	4.5 ± 7.9	10.3 ± 35.5
miR-142-5p	0.5077	>0.9999	>0.9999	>0.9999	1.9 ± 1.0	12.1 ± 15.9	21.3 ± 51.0
miR-130a-3p	0.8636	>0.9999	>0.9999	>0.9999	248.3 ± 653.1	60.9 ± 136.7	122.3 ± 410.9
miR-331-5p	0.5376	>0.9999	>0.9999	0.7997	3.7 ± 1.3	2.4 ± 5.7	5.1 ± 2.0

**Table 3 ijms-25-03363-t003:** *p* value related to Mann–Whitney analysis showing the general rank *p* value and significance of statistical difference between samples with High percentages of *BCR::ABL1* transcript (1% or higher percentage of *BCR::ABL1* transcript) and Health controls. s: significant; ns: non-significant.

miRNA	General Rank (*p* Value)		Mean ± Deviation	
	Mann-Whitney	Significance	High	HC
miR-7-5p	0.2475	ns	1.6 ± 1.7	0.6 ± 0.3
miR-19b-3p	0.9118	ns	10.1 ± 18.2	3.3 ± 3.1
miR-20a-5p	0.8534	ns	2.9 ± 5.0	1.2 ± 0.9
miR-21-5p	0.6842	ns	7.9 ± 13.2	2.6 ± 2.3
miR-25-3p	0.9118	ns	1.9 ± 3.7	0.8 ± 0.7
miR-27b-3p	0.8534	ns	13.7 ± 24.4	4.3 ± 3.8
miR-29a-3p	0.9705	ns	7.6 ± 12.5	2.2 ± 1.7
miR-29b-3p	>0.9999	ns	6.4 ± 10.8	1.8 ± 1.3
miR-92a-1-5p	0.0220	s	3.2 ± 2.5	0.7 ± 0.5
miR-103a-3p	0.7394	ns	4.2 ± 7.4	1.2 ± 1.4
miR-106b-5p	0.9705	ns	3.3 ± 5.4	1.5 ± 1.2
miR-122-5p	0.4813	ns	1.4 ± 2.2	2.4 ± 3.6
miR-125b-5p	0.6842	ns	2.0 ± 3.5	1.2 ± 1.2
miR-130a-5p	No amplification	ns	-	-
miR-150-5p	0.6305	ns	3.7 ± 9.6	0.8 ± 0.5
miR-155-5p	0.4173	ns	42.3 ± 99.2	30.2 ± 45.9
miR-186-5p	0.5787	ns	4.3 ± 7.7	1.9 ± 1.8
miR-192-5p	0.5787	ns	3.2 ± 3.9	1.5 ± 1.0
miR-193b-3p	0.6048	ns	1.3 ± 1.6	0.9 ± 1.0
miR-205-5p	0.2176	ns	5.5 ± 6.2	2.1 ± 1.1
miR-214-5p	No amplification	ns	-	-
miR-221-3p	0.7394	ns	6.7 ± 10.7	1.8 ± 1.5
miR-221-5p	0.8125	ns	24.2 ± 51.1	5.3 ± 6.6
miR-361-5p	0.4359	ns	6.8 ± 10.3	2.6 ± 3.2
miR-451	>0.9999	ns	1.9 ± 3.2	0.7 ± 0.4
miR-486-5p	0.6305	ns	2.1 ± 3.7	0.8 ± 0.6
miR-494-3p	0.8286	ns	36.6 ± 72.0	16.1 ± 17.4

**Table 4 ijms-25-03363-t004:** *p*-values found in Mann–Whitney analysis showing the general rank *p* value and significance of statistical difference between samples with High percentages of *BCR::ABL1* transcript (0.1% or higher percentage of *BCR::ABL1* transcript) and Healthy controls. s: significant; ns: non- significant.

miRNA	General Rank		Mean ± Deviation	
	Mann-Whitney	Significance	High	HC
miR-92a-1-5p	0.057	ns	909.1 ± 4049.7	1.7 ± 5.1
miR-7-5p	0.0353	s	2.6 ± 3.1	9.5 ± 39.2
miR-486-5p	0.0829	ns	3.6 ± 5.8	6.2 ± 23.8
miR-361-5p	0.1494	ns	16.4 ± 36.4	7.4 ± 19.7

**Table 5 ijms-25-03363-t005:** *p* value related to Kruskall–Wallis analysis showing the general rank *p* value and the *p* value for multiple comparisons analyzing samples of buffy coat. High: 0.1% or higher percentage of *BCR::ABL1* transcript; ND: Non-Detected *BCR::ABL1* transcript; HC: Healthy Control group.

miRNA	General Rank	Multiple Comparisons			Mean ± Deviation		
		High vs. ND	High vs. HC	ND vs. HC	High	ND	HC
miR-7-5p	0.7897	>0.9999	>0.9999	>0.9999	3.3 ± 9.3	0.9 ± 1.2	0.5 ± 0.4

**Table 6 ijms-25-03363-t006:** Analysis of patients undergoing continuous follow-up to evaluate the progression of the pathology by evaluating the expression of microRNA-7-5p, expression of the *BCR::ABL1* transcript, treatment, and treatment time. *BCR::ABL1*: Percentage of *BCR::ABL1* transcript in the sample; FC: Fold Change.

	Months of Treatment		*BCR::ABL1* %	FC miR-7-5p	Treatment
Patient 1	0	Point 1	81.74	1	Imatinib
	4	Point 2	13.68	15.637	Imatinib
	12	Point 3	1.854	0.282	Imatinib
	14	Point 4	5.291	38.241	Imatinib
	5	Point 5	0	93.641	Dasatinib
Patient 2	3	Point 1	8.39	1	Imatinib
	6	Point 2	0.000	0.003	Imatinib
	11	Point 3	0.003	0.013	Imatinib
	15	Point 4	0	0.068	Imatinib
Patient 3	8	Point 1	1.943	1	Imatinib
	18	Point 2	0.179	3.985	Imatinib
	21	Point 3	0.268	1.624	Imatinib
	23	Point 4	0.364	23.589	Imatinib
	26	Point 5	2.774	33.697	Imatinib
	27	Point 6	0.96	11.446	Imatinib
Patient 4	0	Point 1	0.143	1	Imatinib
	6	Point 2	72.738	0.020	Imatinib
	9	Point 3	80.616	1.290	Imatinib
	3	Point 4	2.239	0.025	Dasatinib
	5	Point 5	8.192	0.071	Dasatinib
Patient 5	0	Point 1	50.64	1	Imatinib
	3	Point 2	9.138	0.208	Imatinib
	6	Point 3	6.288	0.447	Imatinib
	15	Point 4	1.202	0.133	Imatinib
	22	Point 5	0.902	2.251	Imatinib
	24	Point 6	1.159	1.028	Imatinib
Patient 6	6	Point 1	4.67	1	Imatinib
	13	Point 2	0.578	0.140	Imatinib
	16	Point 3	0.141	1.188	Imatinib
	20	Point 4	0.033	0.733	Imatinib
	29	Point 5	0.000	1.002	Imatinib

**Table 7 ijms-25-03363-t007:** Results of follow-up analyses of patients undergoing free remission treatment, comparing TFR (Treatment Free-Remission) Time and Fold-Change of microRNA-7 (FC miR-7-5p).

		Time	FC miR-7-5p
Patient 1	Point 1	Pause	1
	Point 2	2/3 months	0.849
	Point 3	5/6 months	2.350
	Point 4	14 months	1.810
	Point 5	20 months	0.682
Patient 2	Point 1	In treatment	1
	Point 2	5/6 months	0.253
	Point 3	11/12 months	0.351
	Point 4	14 months	0.410
Patient 3	Point 1	In treatment	1
	Point 2	1 month	0.150
	Point 3	2/3 months	0.200
	Point 4	5/6 months	0.505
	Point 5	7–10 months	0.299
	Point 6	11/12 months	0.458
	Point 7	14 months	0.504
Patient 4	Point 1	In treatment	1
	Point 2	1 months	0.196
	Point 3	2/3 months	0.282
	Point 4	5/6 months	0.462
	Point 5	5/6 months	0.248
	Point 6	5/6 months	0.209
	Point 7	7–10 months	0.222
	Point 8	7–10 months	1.472
	Point 9	7–10 months	0.027
	Point 10	11/12 months	0.100

## Data Availability

The data presented in this study are available on request from the corresponding author.

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
