# Peer review of "The Screening of microRNAs in Chronic Myeloid Leukemia: A Clinical Evaluation"

_ijms, 2024, doi:10.3390/ijms25063363_

Round 1
Reviewer 1 Report
Comments and Suggestions for Authors
Dear authors!
The text of your article is very interesting, but the Tables and Figure 6 raise significant questions.
It would be desirable to add significance levels of differences between groups to Table 1, or to indicate them in the text. In Tables 2-5 it would be desirable to see the values of the studied parameters in the studied groups (in the form of mean±deviation or median and quartiles), not just the almost identical p values. In addition, please correct the typo "High vc. ND" in Tables 2 and 5.
Finally, and most importantly. Figure 6 is completely incorrect. First, the data do not agree with Table 7 (the signature indicates Table 6, but this is obviously wrong neither by number of patients nor by time points); second, Point 1 instead of the same value for all patients, somehow shows values corresponding to the patient number, third - how can you take different time intervals for the same point on the graph? For example, point 3 for different patients turns out to be 2/3 months, 5/6 months, 11/12 months, and even 50 months! I dare to show how the graph in Figure 6 should look like according to the data in Table 7.
Because of the above remarks, I think that the article should be reconsider after major revision. Also pay attention to typos, for example "pexpression", "Sparman's" (this is why I think that Minor editing of English language is required).

Author Response
Reviewer 1 comments:
Dear authors!
The text of your article is very interesting, but the Tables and Figure 6 raise significant questions.
It would be desirable to add significance levels of differences between groups to Table 1, or to indicate them in the text. In Tables 2-5 it would be desirable to see the values of the studied parameters in the studied groups (in the form of mean±deviation or median and quartiles), not just the almost identical p values. In addition, please correct the typo "High vc. ND" in Tables 2 and 5.
Finally, and most importantly. Figure 6 is completely incorrect. First, the data do not agree with Table 7 (the signature indicates Table 6, but this is obviously wrong neither by number of patients nor by time points); second, Point 1 instead of the same value for all patients, somehow shows values corresponding to the patient number, third - how can you take different time intervals for the same point on the graph? For example, point 3 for different patients turns out to be 2/3 months, 5/6 months, 11/12 months, and even 50 months! I dare to show how the graph in Figure 6 should look like according to the data in Table 7.
Because of the above remarks, I think that the article should be reconsider after major revision. Also pay attention to typos, for example "pexpression", "Sparman's" (this is why I think that Minor editing of English language is required).
Corrections:
All the changes in the text are in yellow color, just the corrected typos are not.
It was corrected in the text, the data and information were added to all the solicitated tables, now all the single points tables have data related to P value, significance and mean + deviation. The typo was corrected in the text.
The figure 6 was not incorrect, but different point of views could appear analysing that figure, thinking about it we decided to take off the figure of the text, we think the interpretation is better just analysing the data in the table then using a graph figure, but we agree that thinking in “months” and not in “points” is the best option. Patient 3 was taking off the analysis because we evaluated that the patient 3 hasn’t the first point before the treatment free-remission, so we think it's not so significative to analysing it. About the repeated “points” like 2/3, 5/6… the same patient collected blood to accompany the free-remission treatment like 3 times during the second and third or fifth and sixth months, so there are more samples during the same period but the samples are different, it is not the same samples.
The text was revised and the typos were corrected.
Reviewer 2 Report
Comments and Suggestions for Authors
In the present study, Wosniaki et al. provide a clinical evaluation of signature microRNAs in CML. The study is novel, well-designed and of significant interest to the readers in the field. In addition, the study may open a new therapeutic window in the treatment of CML via targeting signature miRNAs. I have the following minor comments:
1. Please discuss the rationale behind choosing the mentioned 8 miRNAs in the first stage of "2.2 Single-point plasma analysis".
2. Figure 1: It would be better to include the graphical representation of a few miRNAs which show non-significant differences.
3. Figure 1: miR-92a-1-5p: The sample size should be increased.
4. In all of the tables remove the "," and include "." for the p value.
5. Figure 4: miR-7-5p: Put 'ns' for non-significant differences among all the groups calculated.
6. Figure 6: Please explain the reason patient 3-5 data behaving differently than 1-2.
Author Response
Reviewer 2 comments:
In the present study, Wosniaki et al. provide a clinical evaluation of signature microRNAs in CML. The study is novel, well-designed and of significant interest to the readers in the field. In addition, the study may open a new therapeutic window in the treatment of CML via targeting signature miRNAs. I have the following minor comments:
- Please discuss the rationale behind choosing the mentioned 8 miRNAs in the first stage of "2.2 Single-point plasma analysis".
- Figure 1: It would be better to include the graphical representation of a few miRNAs which show non-significant differences.
- Figure 1: miR-92a-1-5p: The sample size should be increased.
- In all of the tables remove the "," and include "." for the p value.
- Figure 4: miR-7-5p: Put 'ns' for non-significant differences among all the groups calculated.
- Figure 6: Please explain the reason patient 3-5 data behaving differently than 1-2.
Corrections:
All the changes in the text are in yellow color, just the corrected typos are not.
- Corrected, we add a short part in the same paragraph explaining that we chosen those microRNAs related to this oncological potential and we add some of the references of this studies.
- About the graphical representation of non-significative results, I think would be worse, we have tried to put some graphs in the first draft but in our conception, we thought that the information looked messy and too much figures, with the tables the information is more organized and readable. We can send you the graphs to you look, but I think the information is better recognized with me tables.
- The sample size was lower in the first screening, but in the next paragraph and in table 4 there is the information about the same microRNA when we evaluated it in a higher number of samples (non-significative, p=0.057).
- Corrected in the text.
- We agree and we putted this information in the legend of the figure 4.
- Each patient has intrinsic differences and we evaluate each patient individually in the respective follow up points. This date shown that other compounds could be interfering in the behaviour of that microRNA in each patient, we explained this during the discussion but we don’t have any sure about the reason responsible for these different behaviours.
Round 2
Reviewer 1 Report
Comments and Suggestions for Authors
Dear authors!
I have read the new version of the manuscript. I support your decision to remove Figure 6. However, there are also comments on the corrected text. The text highlighted in yellow does not look like English in some places ("showing de significances" , "Significative" instead of "significant"). Also, if you write Mean ± deviation (not + ) in the header of the tables, the values should look like "12.1 ± 37.1" (not 12.1 (37.1) ). But in the case of such a large deviation, wouldn't it be better to use medians and quartiles? Here we clearly see a statistical distribution different from normal distribution.
Thus, I think that the manuscript can be accepted after minor revision
Comments on the Quality of English LanguageThe text highlighted in yellow does not look like English in some places ("showing de significances" , "Significative" instead of "significant").
Author Response
Greetings dear reviewer, we made the changes in details that you already cited, the last changes are still in yellow and the new ones were corrected in the text. About mean and deviation or median and quartiles, we tried it and in our opinion the large distribuition is continually appearantly in both modes, so, in our vision the best one was showing the mean and deviation.
Thank you for the attention reading our papper, please, if you find anything else to correct ou change, we will try to reply as soon as possible.
